# Subclinical Carotid Disease Is Associated with Low Serum Vitamin D in Nondiabetic Middle-Aged Hypertensive Patients

**DOI:** 10.3390/nu17030480

**Published:** 2025-01-28

**Authors:** Luca Bulfone, Antonio Vacca, Gabriele Brosolo, Andrea Da Porto, Nicole Bertin, Cinzia Vivarelli, Cristiana Catena, Leonardo A. Sechi

**Affiliations:** 1Clinica Medica, Department of Medicine, University of Udine, 33100 Udine, Italy; luca.bulfone1@gmail.com (L.B.); antonio.vacca94@gmail.com (A.V.); gabriele.brosolo@uniud.it (G.B.); andrea.daporto@uniud.it (A.D.P.); nicole.bertin@uniud.it (N.B.); cinzia.vivarelli@asufc.sanita.fvg.it (C.V.); cristiana.catena@uniud.it (C.C.); 2Hypertension European Excellence Center, Department of Medicine, University of Udine, 33100 Udine, Italy; 3Diabetes Unit, Department of Medicine, University of Udine, 33100 Udine, Italy; 4Thrombosis and Hemostasis Unit, Department of Medicine, University of Udine, 33100 Udine, Italy

**Keywords:** carotid distensibility, carotid plaques, carotid stiffness, hypertension, intima-media thickness, vitamin D

## Abstract

Subclinical carotid artery disease anticipates major cardiovascular events, and previous studies show that low vitamin D levels are associated with arterial stiffening in hypertension. The aim of the study was to examine the relationship of 25-hydroxyvitamin D [25(OH)D] levels with subclinical carotid disease in hypertensive patients. In 223 middle-aged, nondiabetic, primary hypertensive patients free of major cardiovascular and renal complications, we measured 25(OH)D and parathyroid hormone (PTH) and assessed subclinical carotid arteries changes by B-mode ultrasonography. The carotid intima-media thickness (IMT) and presence of plaques were assessed together with measurements of indexes of carotid artery distensibility (coefficient of distensibility) or stiffening (Young’s elastic modulus; β-stiffness). Lower 25(OH)D levels were associated with older age (*p* < 0.001), longer duration of hypertension (*p* = 0.019), higher fasting plasma glucose (*p* = 0.037), and insulin (*p* = 0.044), Homeostatic Model Assessment (HOMA) index (*p* = 0.044), and PTH (*p* < 0.001). Insufficient and deficient 25(OH)D were associated with progressively greater carotid IMT (*p* < 0.001), frequency of carotid plaques (*p* = 0.026), Young’s elastic modulus (*p* = 0.002), and β-stiffness (*p* < 0.001), and progressively lower carotid coefficient of distensibility (*p* < 0.001). Serum levels of 25(OH)D were negatively correlated with age (*p* < 0.001), duration of hypertension (*p* = 0.006), fasting glucose (*p* < 0.001), HOMA index (*p* = 0.032), PTH (*p* < 0.001), carotid IMT (*p* < 0.001), Young’s elastic modulus (*p* = 0.025), and β-stiffness (*p* < 0.001), and positively related with carotid coefficient of distensibility (*p* < 0.001). Multivariate regression analysis showed that both higher carotid IMT (*p* = 0.004) and lower coefficient of distensibility (*p* = 0.002) were related to lower 25(OH)D independent of age, severity, and duration of hypertension and metabolic variables. In conclusion, deficiency/insufficiency of 25(OH)D independently predicts subclinical carotid disease in uncomplicated, middle-aged, hypertensive patients and might predispose these patients to major cardiovascular complications.

## 1. Introduction

Due to its elevated prevalence in the general population, arterial hypertension is retained as the major correctable cardiovascular risk factor and the primary cause of overall worldwide mortality [1]. In hypertension, major cardiovascular events are preceded by subclinical cardiac and vascular changes whose prevalence is constantly rising because of population aging and an increase in exposure to incorrect lifestyles, primarily related to unhealthy dietary habits. Vitamin D insufficiency and deficiency are relatively common conditions [2], and rapidly increasing evidence from the general population [3] and patients with high blood pressure (BP) [4] indicates the existence of a relationship between low 25-hydroxyvitamin D [25(OH)D] levels with cardiovascular events.

As reported in many longitudinal studies, the presence of atherosclerotic plaques in the carotid arteries predicts major cardiovascular events [5,6,7]. In hypertension, early changes in carotid arteries are characterized by subclinical structural and/or functional abnormalities that, much like carotid plaques, provide important information for the stratification of cardiovascular risk [8,9]. Initial structural changes in carotid vessels are identified by ultrasound examination with measurement of the inner layer of the vascular wall, commonly identified as the intima-media thickness (IMT). Mechanical changes occurring in the arterial tree are characterized by wall stiffening that is also identified by ultrasound examination of the carotid arteries or, alternatively, by measurement of the carotid–femoral pulse wave velocity (PWV). Identification of factors that, in addition to major risk factors such as age, overweight/obesity, hyperglycemia, dyslipidemia, and smoking, contribute to carotid IMT thickening and stiffening in hypertension is crucial.

In hypertensive patients, deficient serum 25(OH)D has been associated with cardiac [10] and arterial [11] changes, suggesting a contribution to the development of subclinical hypertension-related cardiovascular damage. In newly diagnosed, treatment-naïve hypertensive patients free of significant comorbidities and major cardiovascular complications, vitamin D deficiency was significantly and independently associated with an increased left ventricular mass [10]. Previous cross-sectional studies investigated the possible relevance of vitamin D status for early arterial changes in small groups of patients with hypertension [12,13,14,15]. Most of these studies were conducted by measuring the carotid–femoral PWV and almost consistently reported an association of deficient vitamin D with stiffening of the arterial tree. On the other hand, findings of studies that investigated the relationships between vitamin D status with IMT in the general population were more controversial [16,17,18,19], and also the only study that included hypertensive patients was inconclusive [20].

Notably, the majority of these previous studies included elderly subjects and subjects with comorbidities, including diabetes and cardiovascular diseases. Because aging is the major contributor to structural and mechanical arterial changes, investigations of younger subjects would be appropriate. Also, the presence of important confounders and comorbidities undermined the relevance of most of these studies.

We are not currently aware of investigations that have examined structural and mechanical changes in carotid arteries in appropriately sized groups of hypertensive patients free of important confounders. Because of the previously suggested importance of 25(OH)D deficiency for subclinical hypertensive arterial damage, we sought to investigate the relationship of 25(OH)D levels with markers of subclinical carotid damage in middle-aged primary hypertensive patients without diabetes and history of major cardiovascular and renal complications.

## 2. Materials and Methods

### 2.1. Patients

Two-hundred-twenty-three consecutive nondiabetic patients with primary hypertension who were referred to the European Excellence Center of Hypertension of the department were included in a cross-sectional study. Subjects referred to the Center are representative of the regional (northeast of Italy) population and include patients with all grades of hypertension [21]. Automated devices (Omron M6; OMRON Healthcare Co., Kyoto, Japan) provided with cuffs of appropriate size were used to measure BP after patients had been seated for at least 15 min, and the mean of three readings was recorded [22]. Hypertension was ascertained with BP measurements performed in at least three separate visits [22]. Patients with age < 18 years or >70 years, body mass index (BMI) > 40 kg/m^2^, secondary hypertension, diabetes mellitus, alcohol abuse, dairy product-free diet, oral vitamin D supplements, 24 h creatinine clearance (GFR) < 60 mL/min per 1.73 m^2^, and history of acute illness in the last 3 months were excluded. Causes of secondary hypertension were excluded according to current guidelines [22] after extensive investigation, as reported previously [23]. People with diabetes were identified according to current guidelines [24] by measurement of fasting glucose (≥126 mg/dL) and glycated hemoglobin (≥6.5%). Anhydrous alcohol consumption was estimated with the use of a comprehensive questionnaire [25], and subjects were considered smokers if they had smoked for more than five years and up to one year before the study. The study was conducted following the principles of the Declaration of Helsinki and was approved by the departmental Institutional Review Board. All patients gave their informed consent.

### 2.2. Laboratory Measurements and Vitamin D Assessment

Venous blood was collected in the morning after an overnight fast with the patient in the sitting position. The glucose-oxidase method was used to measure glucose, and radioimmunoassay to measure insulin. The Homeostatic Model Assessment (HOMA) index was used to assess sensitivity to insulin with the formula [(glucose mmol/L × insulin µU/mL)/22.5] [26]. Chemiluminescence was used to measure serum PTH (Siemens Healthcare srl, Erlangen, Germany; limit of detection: 2.5 pg/mL; intra-assay coefficient of variation of 5.2%) and 25(OH)D (Immunodiagnostic System, Spello, Italy; limit of detection: 3.6 ng/mL; intra-assay coefficient of variation of 5.5%) [27]. Serum vitamin D was considered normal when 25(OH)D level was ≥30 ng/mL (726 nmol/L), insufficient when 25(OH)D level was from 21 to 29 ng/mL (525–725 nmol/L), and deficient when 25(OH)D level was ≤20 ng/mL (50 nmol/L) [28]. Leisure outdoor activity was assessed by standard interviews and subjects were considered physically active when they practiced outdoor activities at least three hours a week. Seasons of 25(OH)D measurement were also considered and defined as spring (March–May), summer (June–August), fall (September–November), and winter (December–February).

### 2.3. Carotid B-Mode Ultrasonography

The left and right carotid arteries were examined with a duplex scanner (Aplio CV, Toshiba, Japan) using a 7 MHz linear array transducer. The same trained operator performed all examinations. Common, internal, and external carotid arteries, and the bifurcation were visualized in the transverse and longitudinal plans for the detection of plaques. Plaques were defined as focal areas of increase in wall thickness of more of the surrounding IMT [29,30]. Carotid IMT was measured with an electronic caliper on the far wall of the distal portion of the common carotid arteries, 1.0 cm proximal to the beginning of the carotid bulb, as described previously [31]. The ultrasound image was frozen in end-diastole by means of ECG-mediated triggering. The mean value of three IMT measurements in a wall segment without plaques was calculated. IMT was considered as the distance between the leading edge of the lumen-intima echo and the leading edge of the media-adventitia echo [32]. Intra-observer variability was 5.2%, and the correlation coefficient of duplicate measurements of IMT was 0.892.

Carotid stiffness was assessed in patients who had been lying quietly for 15 min, and BP was measured in the left arm immediately before performing a carotid artery examination. The pulse pressure (PP) was calculated, and the mean of the last 2 of 3 measurements obtained in 5 min was considered. A longitudinal scan of left and right common arteries was performed under an ECG guide to obtain carotid systolic (SD) and diastolic (DD) diameter, and the mean of measurements obtained in 3 consecutive cardiac cycles was calculated for both arteries according to consensus [33], as reported previously [34]. Three indices of carotid distensibility were calculated as follows:
• *Coefficient of distensibility* (Distensibility)2 × (SD-DD)/DD/pulse pressure (10^−3^/kPa)• *Young’s elastic modulus* (Young)DD/(IMT × Distensibility) (10^3^/kPa)• *Beta-stiffness* (β-stiffness)Ln (systolic BP/diastolic BP)/[(SD-DD)/DD]

Intra-observer variability of all measurements was below 8%, and the correlation coefficient of duplicate measurements for all parameters was >0.850.

### 2.4. Statistical Analysis

The sample size was calculated to provide power in the detection of a 20% difference in IMT and carotid distensibility variables between individuals with normal or deficient 25(OH)D levels with a probability < 5%. Variables with normal distribution are reported as means ± standard deviation (SD). Skewed variables are reported as median and interquartile ranges (IQR) and were log-transformed for analysis. Frequency distributions were compared by Pearson’s chi-square test. Comparisons of 2 groups were performed with Student’s *t* test and comparisons among more than 2 groups with one-way analysis of variance (ANOVA). All comparisons were adjusted for covariates and multiple comparisons with the Bonferroni correction. The relationship between continuously distributed variables was analyzed by linear regression, and the correlation was expressed by Pearson’s coefficient *r*. Multivariate regression was performed to identify variables that were independently related to the carotid variables that were associated with serum 25(OH)D levels in the univariate analysis. In the model, variables were entered sequentially according to the strength of the significance observed in the univariate analysis. Probability (*p*) values of <5% indicated statistical significance. All analyses were performed by using Stata 12.1 (StataCorp LP, College Station, TX, USA).

## 3. Results

The analysis included 223 individuals with primary hypertension (120 males, 103 females; age 50 ± 13 years). Ninety-four (42%) of the 223 patients had never received anti-hypertensive treatment. The characteristics and blood biochemistries of subjects included in the study patients are shown in Table 1, where subjects are divided into three groups according to serum 25(OH)D levels (normality, insufficiency, deficiency). Normal 25(OH)D levels were observed in 92 patients (41%), while of the remaining 131 (59%), 51 (23%) had vitamin D insufficiency and 80 (36%) vitamin D deficiency. Lower 25(OH)D levels were associated with significantly older age, longer duration of hypertension, greater fasting plasma glucose and insulin, HOMA index, and plasma PTH. Sex distribution, BMI, systolic and diastolic BP, use of antihypertensive medications, alcohol consumption, smoking habit, physical exercise, GFR, and levels of total, HDL, and LDL cholesterol, triglycerides, renin, aldosterone, sodium, potassium, calcium, phosphate, and magnesium did not differ among groups.

Analysis of data stratified by age showed progressively lower 25(OH)D with increasing age (18–40 years, 34.8 ± 22.8 mg/mL; 41–55 years, 31.5 ± 19.0; 56–70 years, 24.7 ± 17.0; *p* = 0.012), progressively higher carotid IMT (18–40 years, 573 ± 130 μm; 41–55 years, 651 ± 135 μm; 56–70 years, 805 ± 175 μm; *p* < 0.001), and progressively worse coefficient of distensibility (18–40 years, 45.1 ± 18.8 10^−3^/kPa; 41–55 years, 38.3 ± 15.5 10^−3^/kPa; 56–70 years, 30.5 ± 12.6 10^−3^/kPa; *p* < 0.001). Analysis of data stratified by sex showed no significant differences between men and women for 25(OH)D (males, 29.2 ± 17.4 mg/mL; females, 30.6 ± 21.7 mg/mL; *p* = 0.594), carotid IMT (males, 689 ± 171 μm; females, 681 ± 174 μm; *p* = 0.730), and coefficient of distensibility (males, 36.9 ± 15.0 10^−3^/kPa; females, 38.0 ± 17.3 10^−3^/kPa; *p* = 0.612).

Seasonal serum values of 25(OH)D are shown in Table 2. As expected, levels were significantly higher during the summer and fall than in the spring and winter, whereas no differences were observed in 1,25(OH)D, PTH, calcium, phosphate, and magnesium. 25(OH)D levels were comparable in males and females, physically active and sedentary subjects, smokers and nonsmokers. Also, no significant differences were observed between subjects who were or were not taking antihypertensive agents, nor among patients who had been treated with different drugs.

Figure 1 illustrates the variables that were obtained by B-mode ultrasonography in patients with normal, insufficient, and deficient 25(OH)D serum levels. Carotid IMT was significantly and progressively higher across 25(OH)D serum groups, and the frequency of detectable carotid plaques was significantly higher in patients with either insufficient or deficient 25(OH)D.

Data on carotid distensibility are shown in Figure 2. The coefficient of distensibility decreased, and Young’s elastic modulus and β-stiffness increased significantly and progressively across groups with decreasing 25(OH)D levels.

Linear regression analysis was used to examine the relationships and univariate correlation of serum 25(OH)D with the other study variables (Table 3). Log 25(OH)D levels were significantly and negatively correlated with age, duration of hypertension, glucose, HOMA index, and serum PTH. No correlations were observed with BMI, systolic and diastolic BP, alcohol consumption, renal function, glycated hemoglobin, fasting insulin, and plasma levels of calcium, phosphate, and magnesium. A significant inverse relationship of 25(OH)D was observed with carotid IMT, Young’s elastic modulus, and β-stiffness, and a highly significant direct correlation was found with a carotid coefficient of distensibility (Table 3; Figure 3).

Carotid IMT was significantly and positively correlated with age, BMI, systolic BP, duration of hypertension, glycated hemoglobin, and cholesterol. Carotid coefficient of distensibility was inversely correlated with age, BMI, systolic BP, duration of hypertension, glucose, HOMA index, and triglycerides. Young’s elastic modulus was positively correlated with age, BMI, systolic BP, duration of hypertension, and glycated hemoglobin. Beta-stiffness was positively correlated with age, BMI, glucose, and HOMA index (Table 4).

Multivariate regression analysis was performed, including carotid IMT and coefficient of distensibility as the dependent variables that were treated in the model as continuous variables (Table 5). Variables that were identified in the univariate analysis as possible confounders were entered according to the strength of the association. Carotid IMT was correlated independently with age, systolic BP, log 25(OH)D and log PTH. Carotid coefficient of distensibility was correlated independently with age, BMI, systolic BP, and log 25(OH)D. No independent influence of seasonal differences in 25(OH)D measurements was found with either carotid IMT or coefficient of distensibility.

## 4. Discussion

This study investigated the possible contribution of 25(OH)D levels to subclinical carotid artery changes in middle-aged, nondiabetic, primary hypertensive patients free of clinically relevant organ damage. The results show that insufficient/deficient serum 25(OH)D levels are associated with a greater frequency of carotid plaques, higher carotid IMT, and evidence of carotid artery stiffening. 25(OH)D levels were significantly and negatively related to carotid IMT and directly with the carotid coefficient of distensibility, independent of age, BMI, BP, and other major confounders that might have contributed to subclinical carotid changes. These observations indicate that subnormal vitamin D levels could contribute to early structural and functional carotid abnormalities in uncomplicated hypertensive patients.

Because the detection of early carotid changes gives crucial information for the stratification of cardiovascular risk in hypertension, assessment of subclinical carotid damage in these patients earns specific clinical relevance. Early hypertension-related changes in carotid vessels are characterized by subtle abnormalities that could be assessed noninvasively by ultrasound examination of carotid arteries [35]. On the one hand, early structural changes consist of the thickening of the innermost layer of the vessel wall, which is commonly identified by the IMT [36] and anticipates the development of carotid plaques. Cross-sectional and longitudinal studies have demonstrated that an elevated IMT is associated with increased cardiovascular risk [37,38]. On the other hand, stiffening of the carotid vessels is an early functional marker of hypertension-related vascular damage that also predicts major cardiovascular events [35,39,40]. Therefore, the identification of conditions that, in addition to other major risk factors, could contribute to subclinical carotid damage becomes crucial. Because of the frequent detection of reduced vitamin D concentrations in hypertension [41], we investigated its relevance for subclinical carotid damage, demonstrating an independent influence of vitamin D status on both structural and functional carotid changes.

The relationship of serum vitamin D levels with carotid IMT was examined in past studies providing inconsistent results [16,17,18,19,20]. Initial observations in community-dwelling elderly subjects in the United States reported that carotid IMT increases with decreasing concentrations of 25(OH)D [16]. This finding was not confirmed in a subsequent population study conducted in Asia [17]. Smaller cross-sectional studies conducted in healthy patients also provided opposite results [18,19], and in 98 hypertensive patients, 25(OH)D levels were negatively correlated with IMT, although the relationship was lost when BP was included in a multivariate analysis [20]. In a cross-sectional analysis of 816 elderly participants in the Multi-Ethnic Study of Atherosclerosis (MESA) study, 25(OH)D levels < 20 ng/mL were not associated with significant differences in the carotid coefficient of distensibility and Young’s elastic modulus [42]. Other cross-sectional investigations examined the stiffening of the entire arterial tree by using carotid–femoral PWV. Increased PWV was associated with 25(OH)D deficiency in the general population, mostly involving elderly subjects [43,44,45], and vitamin D supplementation for 8 weeks significantly decreased PWV in an elderly community-based population [46]. Similarly, cross-sectional studies have shown higher PWV in association with vitamin D deficiency in hypertensive patients [9,13,14,15], an association that was more evident in patients with non-dipping hypertension [9]. Most of these studies were conducted in elderly subjects (60 years or more) and aging, as also confirmed by our present data, is the main factor related to both IMT and arterial stiffening. Our study population was purposedly much younger (mean age 50 years), and therefore, the effect of aging on outcome variables was reasonably minimized. Also, most studies included relevant proportions of patients with diabetes and/or previous cardiovascular or renal events, conditions that could impact subclinical carotid disease, and were excluded from our study. Another important confounder that might have affected the results of previous studies explaining some of the inconsistencies of findings is the seasonal changes of 25(OH)D levels because of variations in sunlight exposure. These changes were taken into account in the multivariate model, demonstrating that the associations of 25(OH)D with carotid IMT and coefficient of distensibility occur independently of seasonal changes. Furthermore, in this study, the association of vitamin D with subclinical carotid changes was independent of sex, BMI, BP, outdoor physical exercise, alcohol consumption and smoking, serum electrolytes, and lipid levels.

Possible mechanisms that might contribute to the observed association between vitamin D insufficiency/deficiency and structural and functional carotid changes could be linked to the broad tissue distribution of vitamin D receptors or to indirect effects that could be a consequence of changes in calcium and PTH. Low serum vitamin D may cause IMT thickening and carotid stiffening because vitamin D inhibits the proliferation and migration of vascular smooth muscle cells [47] and macrophage activation with decreased cholesterol uptake and foam cell formation [48]. Moreover, vitamin D decreases T-cell proliferation and the generation of inflammatory cytokines [49] that are directly involved in atherogenesis and block inflammation-triggered endothelial activation and the expression of adhesion molecules. Vitamin D may also act against endothelial dysfunction by inhibition of oxidative molecular and lipid peroxidation and protect from vascular calcification by inhibition of bone morphogenic proteins. Another postulated mechanism is that vitamin D deficiency may activate the renin–angiotensin–aldosterone system that might, in turn, cause arterial changes. However, in our patients, no significant differences in renin and aldosterone levels were detected across groups with different 25(OH)D levels.

This is the first study to examine the association between serum vitamin D levels and subclinical structural and functional carotid changes in subjects with uncomplicated hypertension. The size of the patient sample, the inclusion of selected middle-aged patients without diabetes and major organ complications, and the consideration of all principal confounders in statistical analysis identify the strengths of the study. Also, a combined assessment of structural and mechanical changes in the carotid arteries of hypertensive patients reinforces the relevance of findings, paving the way for possible clinical applications.

Limitations should be highlighted. First, the inclusion of a clinical sample recruited at a tertiary center limits the possibility of extending these results to a larger population. Second, the cross-sectional design of the study obviously limits its potential for causal inference, and therefore, the present results should not lead to conclusively establishing a causal link between low serum levels of 25(OH)D and subclinical carotid disease. Although the robustness of the association and independence from all main potential confounders would suggest causality, longitudinal intervention studies with vitamin D supplementation would be needed to conclusively establish the causal link between vitamin D insufficiency/deficiency and subclinical, hypertension-related, carotid disease. Third, treatment with anti-hypertensive agents was present in 58% of patients included in the study, which could have influenced results. However, we did not observe any significant differences in either 25(OH)D levels or variables of subclinical carotid artery changes between treated and untreated hypertensive patients, nor detect differences among patients who were treated with different types of antihypertensive drugs. Another aspect that deserves full consideration and might limit the potential relevance of the present results is related to the well-known seasonal variations in vitamin D status and region-specific differences due to different sun exposure. As expected, the serum 25(OH)D levels of our patients were significantly higher in the summer and fall than in the winter and spring. Notably, the inclusion of seasonal differences in the multivariate analysis showed that the relationship between 25(OH)D and both carotid IMT and distensibility was independent of these differences. Lastly, all patients lived in the same area in the northeast of Italy, and therefore, no substantial differences in sun exposure might have occurred. In this regard, we have also assessed the time of outdoor physical activity that was comparable across serum vitamin D groups.

## 5. Conclusions

Subclinical changes in target organs predispose subjects with high blood pressure to major cardiovascular events. Several conditions, in addition to elevated BP, could contribute to the development and progression of hypertension-related vascular damage. Low 25(OH)D levels are frequently found in subjects with hypertension and could contribute to subclinical carotid artery changes. This is the first study to show that low 25(OH)D levels are independently associated with both structural and functional carotid changes in a group of middle-aged, nondiabetic, and uncomplicated hypertensive subjects.

Although this study carries limitations in establishing a causal link between low vitamin D status and subclinical carotid changes, the potential clinical implications of these findings deserve some consideration. Because of the robustness of the relationship between low serum 25(OH)D and subclinical carotid disease, serum vitamin D status could be tested in hypertensive subjects as a tool for the timely identification of subclinical carotid disease and a more precise definition of cardiovascular risk. Future studies will be needed to ascertain the possible clinical relevance of the present findings by testing the effects of vitamin D supplementation in hypertensive subjects with low serum 25(OH)D.

## Figures and Tables

**Figure 1 nutrients-17-00480-f001:**
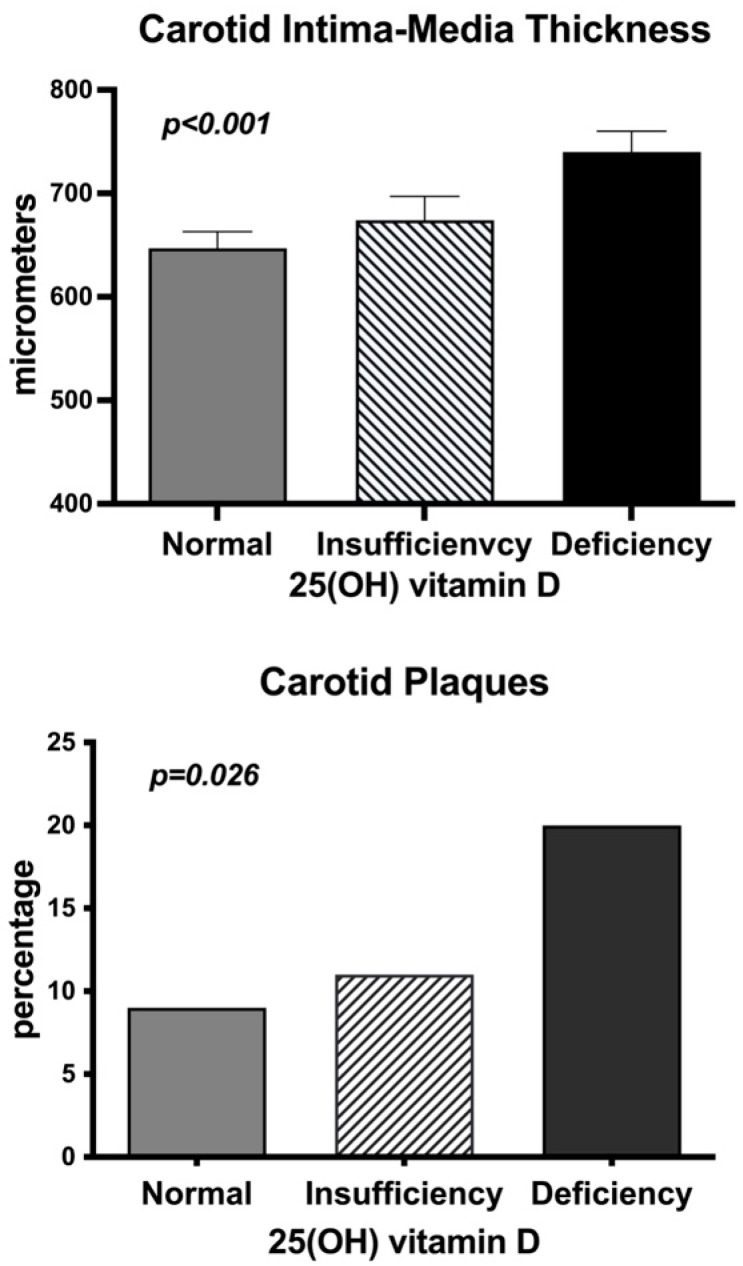
Bar graph showing carotid intima-media thickness values (top panel) and prevalence of carotid plaques in hypertensive patients with normal, insufficient, and deficient 25(OH) vitamin D serum levels. Serum vitamin D was considered normal when 25(OH)D level was ≥30 ng/mL, insufficient when 25(OH)D level was from 21 to 29 ng/mL, and deficient when 25(OH)D level was ≤20 ng/mL.

**Figure 2 nutrients-17-00480-f002:**
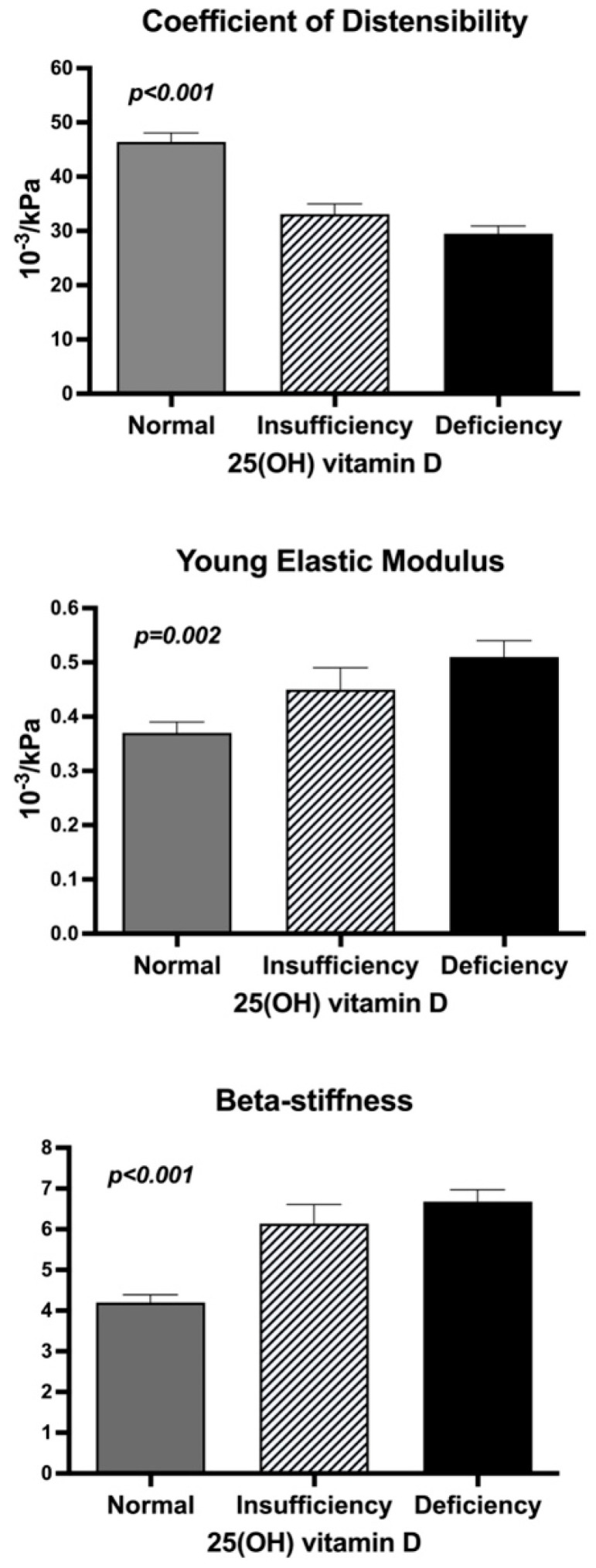
Bar graph showing indexes of carotid distensibility in hypertensive patients with normal, insufficient, and deficient 25(OH) vitamin D serum levels. Serum vitamin D was considered normal when 25(OH)D level was ≥30 ng/mL, insufficient when 25(OH)D level was from 21 to 29 ng/mL, and deficient when 25(OH)D level was ≤20 ng/mL.

**Figure 3 nutrients-17-00480-f003:**
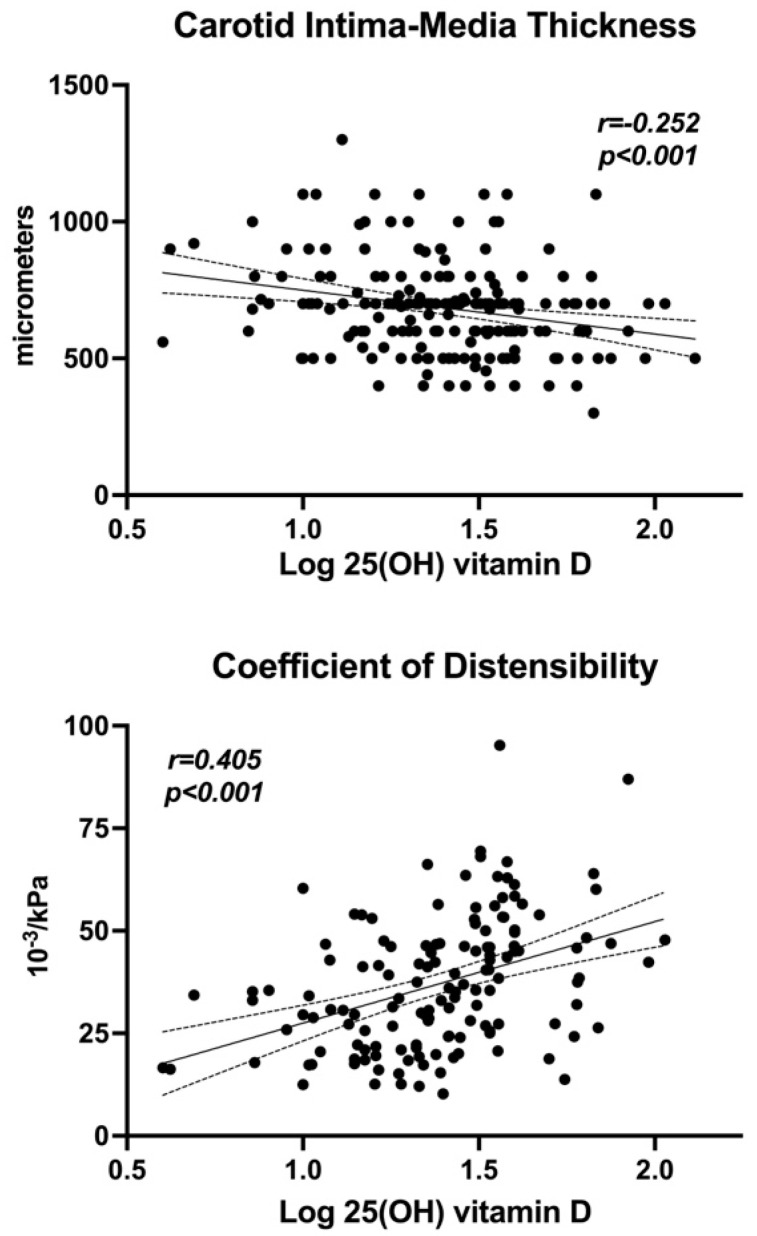
Correlation graph of log 25(OH)D with carotid intima-media thickness (**top panel**) and coefficient of distensibility (**bottom panel**).

**Table 1 nutrients-17-00480-t001:** Characteristics of the study patients according to levels of 25(OH) vitamin D.

Variables	All Patients (*n* = 223)	25(OH)D ≥30 ng/mL (*n* = 92)	25(OH)D 21–29 ng/mL (*n* = 51)	25(OH)D <21 ng/mL (*n* = 80)	*p* Value
** *Clinical characteristics* **					
Age, y	50 ± 13	46 ± 12	52 ± 14	53 ± 12	0.001
Males, *n* (%)	120 (54)	47 (51)	31 (61)	42 (53)	0.515
BMI, kg/m^2^	27.5 ± 4.7	27.0 ± 5.0	27.5 ± 4.3	28.4 ± 4.8	0.158
Systolic BP, mm Hg	149 ± 18	146 ± 18	148 ± 19	152 ± 17	0.089
Diastolic BP, mm Hg	92 ± 13	92 ± 11	90 ± 12	93 ± 14	0.400
Duration of hypertension, y	8 ± 9	6 ± 8	8 ± 8	10 ± 11	0.019
Antihypertensive Tx, *n* (%)	129 (58)	46 (50)	31 (61)	52 (65)	0.124
Alcohol consumption, g/day	9 ± 18	8 ± 16	11 ± 23	8 ± 14	0.551
Smokers, *n* (%)	52 (23)	21 (23)	12 (24)	19 (24)	0.989
Physically active, *n* (%)	53 (24)	22 (24)	16 (31)	15 (19)	0.254
Season					
Spring, *n* (%)	65 (29)	22 (24)	15 (29)	28 (35)	0.280
Summer, *n* (%)	30 (13)	20 (22)	6 (12)	4 (5)	0.005
Fall, *n* (%)	58 (26)	27 (29)	10 (20)	21 (26)	0.445
Winter, *n* (%)	70 (31)	23 (25)	20 (39)	27 (34)	0.183
Summer/autumn	88 (39)	47 (51)	16 (31)	25 (31)	0.012
** *Blood biochemistries* **					
Creatinine, mg/dL	0.92 ± 0.22	0.93 ± 0.17	0.89 ± 0.17	0.92 ± 0.29	0.579
GFR, mL/min·1.73 m^2^	100 ± 27	102 ± 27	101 ± 25	97 ± 29	0.467
Sodium, mmol/L	141 ± 2	141 ± 2	141 ± 3	141 ± 3	1.000
Potassium, mmol/L	4.07 ± 0.38	4.04 ± 0.35	4.11 ± 0.41	4.07 ± 0.40	0.576
Glucose, mg/dL	91 ± 13	89 ± 9	92 ± 12	93 ± 11	0.037
Glycated hemoglobin, %	5.6 ± 0.6	5.5 ± 0.4	5.6 ± 0.5	5.6 ± 0.8	0.466
Insulin, µUI/mL	7.3 (4.1–11.5)	6.3 (4.0–10.0)	6.8 (3.5–11.6)	8.8 (49–12.0)	0.044
HOMA index	1.60 (0.85–2.62)	1.33 (0.84–1.99)	1.51 (0.73–2.78)	1.95 (1.13–3.01)	0.044
Triglycerides, mg/dL	117 ± 66	105 ± 55	120 ± 63	129 ± 77	0.055
Cholesterol, mg/dL	198 ± 41	197 ± 41	196 ± 43	201 ± 39	0.739
HDL cholesterol, mg/dL	57 ± 19	60 ± 18	56 ± 16	55 ± 21	0.188
LDL cholesterol, mg/dL	118 ± 36	115 ± 34	115 ± 37	122 ± 37	0.376
Renin, mUI/mL	9.7 (5.0–19.8)	10.9 (5.9–19.7)	8.9 (6.2–18.0)	8.9 (4.6–19.4)	0.771
Aldosterone, pg/mL	123 ± 78	123 ± 82	109 ± 72	131 ± 77	0.296
25(OH)D, mg/mL	25.4 (16.4–36.0)	38.0 (34.0–55.1)	24.5 (22.4–26.0)	14.4 (10.6–17.7)	<0.001
1,25(OH)D, pg/mL	59 (43–95.0)	79 (53–128)	52 (36–64)	50 (34–67)	<0.001
PTH, pg/mL	62 (44–81)	53 (39–69)	60 (47–77)	73 (60–101)	<0.001
Calcium, mg/dL	9.2 ± 0.5	9.1 ± 0.5 (60–101)	9.2 ± 0.5	9.2 ± 0.4	0.310
Phosphate, mmol/L	1.04 ± 0.26	1.05 ± 0.24	1.05 ± 0.19	1.04 ± 0.31	0.962
Magnesium, mmol/L	0.86 ± 0.17	0.86 ± 0.15	0.84 ± 0.08	0.87 ± 0.23	0.623

Data are shown as mean ± standard deviation or as median [interquartile range]. Comparisons between individuals with different serum 25(OH)D levels were performed by analysis of variance. Variables with skewed distribution were log-transformed. Comparison of frequency distributions was performed with the Pearson chi-square test. Reported *p* values for the comparison among patients with normal, insufficient, and deficient 25(OH)D serum levels. To convert to international units, multiply glucose by 0.05551 (mmol/L), insulin by 7.175 (pmol/L), and calcium by 0.2495 (mmol/L). Abbreviations: BMI, body mass index; GFR, glomerular filtration rate; HOMA, homeostatic model assessment; HDL, high-density lipoprotein; LDL, low-density lipoprotein; 25(OH)D, 25-hydroxyvitamin D; 1,25(OH)D, 1,25-hydroxyvitamin D; PTH, parathyroid hormone.

**Table 2 nutrients-17-00480-t002:** Vitamin D and related variables that were measured during different seasons.

Variables	Spring (*n* = 65)	Summer (*n* = 30)	Fall (*n* = 58)	Winter (*n* = 70)	*p* Value
** *Blood biochemistries* **					
25(OH)D, mg/mL	22.4 (11.2–33.5)	33.6 (24.9–54.8)	28.0 (18.9–78.0)	22.7 (15.0–33.4)	<0.001
1,25(OH)D, pg/mL	57 (43–77)	67 (62–93)	58 (39–109)	53 (40–97)	0.281
PTH, pg/mL	67 (56–84)	55 (44–68)	59 (41–72)	64 (43–81)	0.154
Calcium, mg/dL	9.3 ± 0.5	9.1 ± 0.5	9.2 ± 0.4	9.2 ± 0.4	0.214
Phosphate, mmol/L	1.09 ± 0.33	0.99 ± 0.15	1.04 ± 0.28	1.03 ± 0.18	0.303
Magnesium, mmol/L	0.88 ± 0.25	0.85 ± 0.05	0.87 ± 0.18	0.83 ± 0.08	0.347

Data are shown as mean ± standard deviation or as median [interquartile range]. Comparisons among seasons were performed by analysis of variance. Variables with skewed distribution were log-transformed. Reported *p* values for the comparison among measurements performed in different seasons. To convert to international units, multiply calcium by 0.2495 (mmol/L). Abbreviations: 25(OH)D, 25-hydroxyvitamin D; 1,25(OH)D, 1,25-hydroxyvitamin D; PTH, parathyroid hormone.

**Table 3 nutrients-17-00480-t003:** Relationships of log 25(OH)D with the study variables.

Variables	*r*	*p* Value		Variables	*r*	*p* Value
Age	−0.248	<0.001		HOMA index	−0.146	0.032
BMI	−0.116	0.090		1-25(OH)D3	0.387	<0.001
Systolic BP	−0.092	0.190		PTH	−0.362	<0.001
Diastolic BP	−0.018	0.797		Calcium	−0.121	0.076
Duration hypertension	−0.186	0.006		Phosphate	0.025	0.724
Alcohol consumption	−0.016	0.819		Magnesium	−0.068	0.331
GFR	−0.008	0.906		Carotid IMT	−0.252	<0.001
Glucose	−0.232	<0.001		Distensibility	0.405	<0.001
Glycated hemoglobin	−0.092	0.237		Young	−0.189	0.025
Insulin	−0.113	0.099		Beta-stiffness	−0.365	<0.001

The relationships between continuously distributed variables were examined with the linear regression analysis, and the correlation expressed by Pearson’s correlation coefficient *r*. Abbreviations: BMI, body mass index; BP, blood pressure; GFR, glomerular filtration rate; HOMA, homeostatic model assessment; PTH, parathyroid hormone; IMT, intima-media thickness.

**Table 4 nutrients-17-00480-t004:** Relationships of carotid indexes with the study variables.

	Carotid IMT		Coefficient of Distensibility		Young’s Elastic Modulus		Beta-Stiffness	
	*r*	*p* Value	*r*	*p* Value	*r*	*p* Value	*r*	*p* Value
** *Clinical characteristics* **								
Age	0.524	<0.001	−0.360	<0.001	0.359	<0.001	0.296	<0.001
BMI	0.001	0.984	−0.280	<0.001	0.196	0.021	0.292	<0.001
Systolic BP	0.242	<0.001	−0.200	<0.018	0.249	0.003	0.077	0.365
Diastolic BP	0.106	0.152	−0.064	0.451	0.055	0.519	0.027	0.754
Duration of hypertension	0.250	<0.001	−0.154	0.038	0.261	0.002	0.126	0.066
Alcohol consumption	0.132	0.077	−0.110	0.132	0.132	0.077	0.069	0.427
** *Biochemical variables* **								
GFR	−0.047	0.514	0.109	0.198	−0.070	0.408	−0.128	0.067
Glucose	0.137	0.053	−0.188	0.025	0.107	0.206	0.190	0.024
Glycated hemoglobin	0.203	0.013	−0.135	0.055	0.192	0.039	0.162	0.053
HOMA index	0.013	0.855	−0.273	0.001	0.110	0.202	0.228	0.007
Triglycerides	0.089	0.213	−0.174	0.039	0.087	0.309	0.165	0.052
Total cholesterol	0.183	0.008	−0.032	0.706	0.099	0.245	0.080	0.343
HDL cholesterol	−0.122	0.092	0.134	0.115	−0.027	0.752	−0.116	0.171
LDL cholesterol	0.099	0.171	−0.048	0.576	0.090	0.289	0.092	0.280

The relationships between continuously distributed variables were examined with the linear regression analysis, and the correlation expressed by Pearson’s correlation coefficient *r*. Abbreviations: IMT, intima-media thickness; BMI, body mass index; BP, blood pressure; GFR, glomerular filtration rate; HOMA, homeostatic model assessment; HDL, high-density lipoprotein; LDL, low-density lipoprotein.

**Table 5 nutrients-17-00480-t005:** Multivariate analysis with carotid intima-media thickness and coefficient of distensibility as the dependent variables.

	Carotid IMT				Coefficient of Distensibility		
	β	*F* Value	*p* Value		β	*F* Value	*p* Value
Age	6.20	35.21	<0.001	Age	−0.34	10.41	0.002
BMI	3.43	2.39	0.124	BMI	−0.73	6.07	0.015
Systolic BP	1.84	7.95	0.006	Systolic BP	−0.15	4.11	0.045
Duration of hypertension	0.53	0.13	0.715	Duration of hypertension	0.25	2.43	0.122
Glycated hemoglobin	3.58	0.02	0.899	Glucose	0.01	0.07	0.993
Total cholesterol	0.23	0.42	0.420	HOMA index	−0.11	0.16	0.677
Log 25(OH)D	−166.8	8.56	0.004	Triglycerides	−0.02	0.91	0.341
Log PTH	−129.7	4.32	0.040	Log 25(OH)D	18.24	10.17	0.002
Summer/Fall	−3.18	1.97	0.091	Log PTH	−7.66	1.31	0.255
				Summer/Fall	0.19	1.87	0.161

Abbreviations: IMT, intima-media thickness, BMI, body mass index; BP, blood pressure; HOMA, homeostatic model assessment; 5(OH)D, 25-hydroxyvitamin D; PTH, parathyroid hormone.

## Data Availability

Data are contained within the article will be fine.

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
