# Peer review of "Subclinical Carotid Disease Is Associated with Low Serum Vitamin D in Nondiabetic Middle-Aged Hypertensive Patients"

_nutrients, 2025, doi:10.3390/nu17030480_

Round 1
Reviewer 1 Report
Comments and Suggestions for Authors
The study addresses an important and clinically relevant topic, exploring the association between vitamin D deficiency and subclinical carotid disease in middle-aged hypertensive patients. The methodology is robust, with clear inclusion/exclusion criteria and comprehensive consideration of confounding factors. However, the cross-sectional design limits causal inference, which should be emphasized further in the discussion. The seasonal variation in vitamin D measurements and its potential bias, along with the region-specific study population, should also be more explicitly acknowledged as limitations. Figures and tables could benefit from more descriptive captions, and terminology like "HOMA-index" should be briefly explained for broader accessibility. The discussion would be strengthened by expanding on potential mechanisms linking vitamin D to vascular changes and including more experimental evidence. Additionally, stratifying results by age and sex, while elaborating on the rationale for the selected population, could enhance insights. Finally, the implications for routine clinical assessment of vitamin D in hypertension should be cautiously discussed, with future directions emphasizing the need for longitudinal and interventional studies to establish causality and therapeutic relevance.
Author Response
Reviewer 1
We thank the Reviewer for his/her thoughtful comments and suggestions that have considerably improved our manuscript. Here are our responses to the points that were raised.
The study addresses an important and clinically relevant topic, exploring the association between vitamin D deficiency and subclinical carotid disease in middle-aged hypertensive patients. The methodology is robust, with clear inclusion/exclusion criteria and comprehensive consideration of confounding factors.
- Thank you.
However, the cross-sectional design limits causal inference, which should be emphasized further in the discussion. The seasonal variation in vitamin D measurements and its potential bias, along with the region-specific study population, should also be more explicitly acknowledged as limitations.
- Limitations due to the cross-sectional design have been further emphasized (lines 358-365) and additional comments on the possible relevance of seasonal and region-specific variations in 25(OH)D levels have been added in the revised manuscript (lines 370-380).
Figures and tables could benefit from more descriptive captions, and terminology like "HOMA-index" should be briefly explained for broader accessibility.
- More descriptive captions and explanation of abbreviations have been added in figures and tables as requested (lines 185-186; 190-193; 211-212; 214-215; 227-229; 233-235; 247-249; 261-264; 275-277). Thank you.
The discussion would be strengthened by expanding on potential mechanisms linking vitamin D to vascular changes and including more experimental evidence.
- We have expanded the discussion of the potential mechanisms that might link vitamin D to subclinical carotid changes in the revised version of the manuscript (lines 339-346). Thank you.
Additionally, stratifying results by age and sex, while elaborating on the rationale for the selected population, could enhance insights.
- Analysis of data stratified by age showed progressively lower 25(OH)D with increasing age (18-40 years, 34.8±22.8 mg/ml; 41-55 years, 31.5±19.0; 56-70 years, 24.7±17.0; p=0.012), progressively higher carotid IMT (18-40 years, 573±130 mm; 41-55 years, 651±135 mm; 56-70 years, 805±175 mm; p<0.001), and progressively worse coefficient of distensibility (18-40 years, 45.1±18.8 10-3/kPa; 41-55 years, 38.3±15.5 10-3/kPa; 56-70 years, 30.5±12.6 10-3/kPa; p<0.001). Analysis of data stratified by sex showed no significant differences between men and women for 25(OH)D (males, 29.2±17.4 mg/ml; females, 30.6±21.7 mg/ml; p=0.594), carotid IMT (males, 689±171 mm; females, 681±174 mm; p=0.730), and coefficient of distensibility (males, 36.9±15.0 10-3/kPa; females, 38.0±17.3 10-3/kPa; p=0.612). New information has been added in the results section of the revised manuscript (lines 194-202).
Finally, the implications for routine clinical assessment of vitamin D in hypertension should be cautiously discussed, with future directions emphasizing the need for longitudinal and interventional studies to establish causality and therapeutic relevance.
- Discussion on possible implications for routine clinical assessment of vitamin D status in hypertension has been reconsidered with more caution as requested (lines 389-394). Additional longitudinal intervention studies with vitamin D supplementation are needed to conclusively establish the causal link between vitamin D insufficiency/deficiency and subclinical hypertension-related carotid disease and to demonstrate the possible clinical relevance of this intervention in hypertension. These points have been emphasized in the discussion of the revised manuscript (lines 363-365 and 394-396).

Reviewer 2 Report
Comments and Suggestions for Authors
Dear Authors,
This study was conducted to subclinical carotid disease is associated with low serum vitamin D in nondiabetic middle-aged hypertensive patients. I would like to thank the authors for their work on this manuscript. In general, the manuscript is quite good. I believe this work was excellent issue in field of nutrition, public health and medicine section.
Abstract
Line 24: HOMA-index à Homeostatic Model Assessment-index (HOMA-index)
Abbreviations are usually defined at the first use
Please add all results of statistical exact p-value in each variables in Results section.
Please sort alphabetically in Key-words.
Introduction
Please could you more explain the purpose of this study for Nondiabetic Middle-Aged Hypertensive Patients. It should be added 4-5 paragraphs for purpose of this study in Nondiabetic Middle-Aged Hypertensive Patients.
Moreover, in general, the Introduction section was too short contents. You have to add some paragraphs that backgrounds or literature reviews between subclinical carotid disease and low serum vitamin D.
Method: well written
Line 135: one-way ANOVA à one-way analysis of variance
Results
All Tables and Figures, you have to insert abbreviation and full name in footnote. For example, Table 1, BMI, HDL, LDL, etc. have to be shown abbreviation and full name in footnote.
All Tables have to be revised SD, IQR, and ANOVA. For example, Table 1,
Line 161: Data are shown as mean ± SD or as median [IQR]. à Data are shown as mean ± standard deviation or as median (interquartile range).
Line 162: done by ANOVA à done by analysis of variance.
Because this manuscript has many results of data, please you should double check all results of data in Results section, again.
Please revise all results to two decimal places in mean, standard deviation, etc., and three decimal places in statistical values (t, F value, p-value) are generally spelled out in academic writing. Please change in whole manuscript and all Tables and Figures.
Discussion
You should add strengths, and application in field of this study. And, you should add more limitations in this study. Furthermore, checking by the iThenticate system, the plagiarism rate was 49% (quotes included and bibliography excluded). I believe that it is not acceptable plagiarism rate. Please reduce the plagiarism rate under 15%.
Author Response
Reviewer 2
We thank the Reviewer for his/her thoughtful comments and suggestions that have considerably improved our manuscript. Here are our responses to the points that were raised.
This study was conducted to subclinical carotid disease is associated with low serum vitamin D in nondiabetic middle-aged hypertensive patients. I would like to thank the authors for their work on this manuscript. In general, the manuscript is quite good. I believe this work was excellent issue in field of nutrition, public health and medicine section.
- Thank you.
Abstract
Line 24: HOMA-index Homeostatic Model Assessment-index (HOMA-index). Abbreviations are usually defined at the first use.
Please add all results of statistical exact p-value in each variables in Results section.
Please sort alphabetically in Key-words.
- Abbreviations have been explained (line 25). Exact p-values of variables included in the abstract have been added (lines 24-33). Key-words are now sorted alphabetically (lines 38-39).
Introduction
Please could you more explain the purpose of this study for Nondiabetic Middle-Aged Hypertensive Patients. It should be added 4-5 paragraphs for purpose of this study in Nondiabetic Middle-Aged Hypertensive Patients. Moreover, in general, the Introduction section was too short contents. You have to add some paragraphs that backgrounds or literature reviews between subclinical carotid disease and low serum vitamin D.
- Previous studies conducted in the general population and in small groups of hypertensive patients included elderly subjects and subjects with comorbidities such as diabetes and cardiovascular diseases. Because aging is the major contributor to structural and mechanical arterial changes investigations of younger subjects would be appropriate. Also, presence of important confounders and comorbidities undermined the relevance of most of these reports together with the lack of consideration of seasonal variations of serum 25(OH)D. This is why we included relatively young nondiabetic hypertensive patients free of important comorbidities and it has been explained in the introduction of the revised manuscript (lines 76-83). Also and as requested, concise reference to the previously existing literature on the subject has been added to the introduction (lines 55-60 and 65-75). Thank you.
Methods well written
Line 135: one-way ANOVA one-way analysis of variance.
- ANOVA has been spelled out (lines 160-161).
Results
All Tables and Figures, you have to insert abbreviation and full name in footnote. For example, Table 1, BMI, HDL, LDL, etc. have to be shown abbreviation and full name in footnote. All Tables have to be revised SD, IQR, and ANOVA. For example, Table 1,
Line 161: Data are shown as mean ± SD or as median [IQR]. Data are shown as mean ± standard deviation or as median (interquartile range).
Line 162: done by ANOVA done by analysis of variance.
- Abbreviations have been spelled out in the footnotes of tables and figures (lines 185-186; 190-193; 211-212; 214-215; 227-229; 233-235; 247-249; 261-264; 275-277).
Because this manuscript has many results of data, please you should double check all results of data in Results section, again.
- All results of data in the Results section have been checked for correctness.
Please revise all results to two decimal places in mean, standard deviation, etc., and three decimal places in statistical values (t, F value, p-value) are generally spelled out in academic writing. Please change in whole manuscript and all Tables and Figures.
- We have revised data and reported values to the least meaningful decimal place. Additional decimal places are omitted for variables such age, blood pressure, alcohol consumption, duration of hypertension etc…because these would be meaningless. All statistical values are reported with three decimals and have been spelled out in academic writing.
Discussion
You should add strengths, and application in field of this study. And, you should add more limitations in this study. Furthermore, checking by the iThenticate system, the plagiarism rate was 49% (quotes included and bibliography excluded). I believe that it is not acceptable plagiarism rate. Please reduce the plagiarism rate under 15%.
- Discussion on the strenghts and application together with limitations of the study has been expanded in the revised version of the manuscript (lines 350-356; lines 357-380; lines 389-396). Text had been already checked for the repetition rate by the editorial office. This was higher that 3% in only 2 papers previously published by our group and this was due to technical parts of text (mostly in methods and tables) that have already been modified trying to preserve the content. We have further revised the manuscript following the same document previously provided by the editorial office and believe that what now remains is due to small groups of words that would be difficult to change. Thank you.

Round 2
Reviewer 2 Report
Comments and Suggestions for Authors
It is very good revision. Thank you!